# Genetic Preference for Sweet Taste in Mothers Associates with Mother-Child Preference and Intake

**DOI:** 10.3390/nu15112565

**Published:** 2023-05-30

**Authors:** Pernilla Lif Holgerson, Pamela Hasslöf, Anders Esberg, Simon Haworth, Magnus Domellöf, Christina E. West, Ingegerd Johansson

**Affiliations:** 1Department of Odontology, Section of Pediatric Dentistry, Faculty of Medicine, Umeå University, SE-90185 Umeå, Sweden; pamela.hasslof@umu.se; 2Department of Odontology, Section of Cariology, Faculty of Medicine, Umeå University, SE-90185 Umeå, Sweden; anders.esberg@umu.se (A.E.); ingegerd.johansson@umu.se (I.J.); 3Bristol Dental School, University of Bristol, Bristol BS8 2BN, UK; simon.haworth@bristol.ac.uk; 4MRC Integrative Epidemiology Unit, Department of Population Health Sciences, Bristol Medical School, University of Bristol, Bristol BS8 2BN, UK; 5Department of Clinical Sciences, Section of Pediatric medicine, Faculty of Medicine, Umeå University, SE-90185 Umeå, Sweden; magnus.domellof@umu.se (M.D.); christina.west@umu.se (C.E.W.)

**Keywords:** mothers with child, NorthPop cohort, single nucleotide polymorphisms, taste, sweet intake, sweet preference

## Abstract

Taste perception is a well-documented driving force in food selection, with variations in, e.g., taste receptor encoding and glucose transporter genes conferring differences in taste sensitivity and food intake. We explored the impact of maternal innate driving forces on sweet taste preference and intake and assessed whether their children differed in their intake of sweet foods or traits related to sweet intake. A total of 133 single nucleotide polymorphisms (SNPs) in genes reported to associate with eating preferences were sequenced from saliva-DNA from 187 mother-and-child pairs. Preference and intake of sweet-, bitter-, sour-, and umami-tasting foods were estimated from questionnaires. A total of 32 SNP variants associated with a preference for sweet taste or intake at a *p*-value < 0.05 in additive, dominant major, or dominant minor allele models, with two passing corrections for multiple testing (q < 0.05). These were rs7513755 in the *TAS1R2* gene and rs34162196 in the *OR10G3* gene. Having the T allele of rs34162196 was associated with higher sweet intake in mothers and their children, along with a higher BMI in mothers. Having the G allele of rs7513755 was associated with a higher preference for sweets in the mothers. The rs34162196 might be a candidate for a genetic score for sweet intake to complement self-reported intakes.

## 1. Introduction

An unhealthy diet increases the risk of many poor health conditions. Dietary factors are the fifth most important risk factor for disability-adjusted life years (DALYs), with more than 10% of DALYs being attributed to an unhealthy diet (starvation excluded) in many countries [1]. The definition of a healthy diet has largely been stable for decades, but despite this, a preference for hedonic over healthy eating remains common in societies with access to today’s huge variety of foods, many of which are less healthy processed foods with high sugar, saturated fat, and salt content. This raises the question of the role of innate driving forces in eating behaviours.

Taste, smell, and appearance are key factors in food preference [2]. Humans can discriminate five basic tastes, i.e., sweet, bitter, sour, salty, and umami. G-protein-mediated signalling through well-characterised chemosensory receptors expressed on the taste buds on the tongue and soft palate initiates the taste sensation [3]. Receptors for recognition of sweet- and umami-tasting ligands are encoded by genes in the *TAS1R* gene family (mainly *TAS1R1, TAS1R2,* and *TAS1R3*), with the dimeric protein from the *TAS1R2/TAS1R3* complex responding to natural and artificial sweet ligands and *TAS1R1/TAS1R3* to umami taste [4]. Proteins encoded by the *TAS2R* gene family confer the perception of bitter compounds. Additionally, salty and acidic tastes are detected from transduction through ion channels, such as the suggested epithelial sodium channel (ENaC) and OTOP1 receptors, respectively [5,6], and receptors like CD36 and GPR120 are involved in fat perception [7]. Other proteins involved in taste mediation or food selection include the gustducin protein (encoded by the *GNAT3* gene), the families of glucose transporters (such as SLC2A3 and SLC2A4), and carbonic anhydrases (CA6) [8,9]. There is natural variation in the genes encoding these proteins, providing one possible explanation for differences in innate taste preference and food selection between people. Variants in the *TAS2R* and *TAS1R* gene families are well known to correlate with bitter, sweet, and umami preferences [10], and recently, a variant in the gene encoding for the olfactory receptor 10G3 protein was reported to differ among fruit consumers in a genome-wide association study (GWAS) nested in the UK Biobank [11]. Alongside roles in taste sensation, encoded proteins are reported to have other functions and to be expressed outside the oral cavity; for example, the *TAS2R*-derived proteins have innate immunity functions [12,13]. Thus, various phenotypical traits have been linked to genetic variants in this type of gene, such as dietary patterns, dental caries, BMI, and cardiometabolic disease [8,14].

Childhood is a period where dietary balance is especially important for the child’s growth and cognitive development [15,16]. A child’s acceptance of certain foods results from a complex interplay between innate (genetic) determinants and external influences. Studies show that infants have a naive preference for sweet and salty tastes and reject sour and bitter tastes, but also a plasticity in both food acceptance and eating behaviour [17]. Here, caregivers play a key role in shaping the dietary habits of their children. For example, Fisk et al. [18] reported that maternal diet traits explained 30% of the variance in the child’s diet quality, emphasising the importance of the mother as a role model, and several studies report associations between the child’s diet quality and maternal diet during pregnancy and lactation [19,20]. If mothers have an innate preference for some food types, e.g., sweet foods, this may affect the foods they select for their children and subsequently shape their eating habits in addition to the child’s own inherited preference for foods or tastes. The primary objective of the present study was to explore the impact of maternal genetic driving forces on sweet preference and reported intake and assess whether children of mothers with a genetic preference for sweet were given more sweet foods and had a higher sucrose intake. Secondary objectives included assessment of associations with other taste variants, BMI, and dental caries in the mothers. A panel of 133 single nucleotide polymorphisms (SNPs) in genes reported to be associated with eating preferences and intake was sequenced in 187 mothers. Preferences and habitual intake of sweet-, bitter-, sour-, and umami-tasting foods were recorded for the mothers and their up to 35 months old children. The hypothesis was that polymorphisms in taste-related genes with an impact on the mother’s preference and intake of sweet foods link to higher consumption of sweet-tasting products and sucrose intake in her child.

## 2. Materials and Methods

### 2.1. Study Participants

The study was carried out as an extension of the prospective, population-based NorthPop Birth Cohort Study recruiting in the larger Umeå area in northern Sweden (https://www.northpop.se/en/home-2/, accessed on 1 February 2023). From late September 2019 to early October 2021, mothers of children in the NorthPop who accompanied their 3 year old child to the dental office for a first dental check-up were consecutively invited to participate in the study by donating a saliva sample for DNA extraction and by responding to a food preference questionnaire. Of the 258 invited mothers, 251 had at least one previous dietary recording in NorthPop, 187 consented to saliva donation, and 142 responded to the food preference questionnaire (Appendix A).

The study is reported in accordance with the STROBE-nut protocol for cross-sectional studies (Appendix A).

### 2.2. Diet Recordings for the Mothers and the Children

Food intake was recorded by a food frequency questionnaire (FFQ) when the mother was in gestational week 35 (FFQ1) and when the child was 24 months old (FFQ2). FFQ1 and FFQ2 encompassed 125 food/food aggregates covering Swedish eating habits. Habitual intake for the previous year was recorded, with solid foods scored on a 5-level scale ranging from rarely/never to 3 times a day or more, and beverages on a 6- or 7-level scale, including rarely/never for glasses or cups per month, week, or day. The mothers also responded to a questionnaire with a list of 41 food items selected to represent the five basic tastes (6 items for sweet, 16 items for bitter, 6 items for sour, 5 items for umami, and additional items for neutral taste). This questionnaire, which has been validated against the dilution series of the respective tastes [8], was distributed to the mothers when the child was 35 months old, and the response options were on a six-level scale ranging from “disgust” to “love”.

Diet intake for the child was recorded when the child was (mean (99% CI)) 19.1 (19.0, 19.2) and 35.3 (35.0, 35.6) months old, respectively. An extensive questionnaire, including an FFQ section, was answered by the accompanying caregiver (mostly the mother). On the first occasion, the FFQ included questions on 39 food items, and at the latter visit, questions on 190 foods were combined with consumed amounts.

The reported FFQ scores were transformed into intakes per week for snacking or treat-type sweet foods and aggregated into scores for the weekly intake frequency of sweet foods for the mothers and children, respectively. For the mothers, corresponding intake frequencies were aggregated for foods with bitter, sour, and umami tastes. For the children, total sucrose intake per week was estimated from their second FFQ by transforming the reported scores to intake frequency per week and weighting by the reported amounts and sucrose content reported in the database at the Swedish Food Agency (https://www7.slv.se/SokNaringsinnehall/, accessed on 1 August 2020). The foods included in the various scores are presented in Appendix A.

### 2.3. Anthropometric Information for the Mothers and the Children

Information about the height and body weight of the mothers was retrieved from the records at the maternity health care unit (measures from her first visit to the ward, i.e., early 1st trimester). BMI was calculated as weight in kg/height in m^2^. Women were classified as normal weight if their BMI was <25 kg/m^2^, overweight if 25–29.9 kg/m^2^, and obese if ≥30 kg/m^2^. The body weight and height of the children were retrieved from the childcare centre when the child was 18 and 36 months old. The child’s body mass index percentile (BMI percentile adjusted for age and sex) was from the Centres for Disease Control and Prevention (CDC) BMI percentile calculator for children (https://www.cdc.gov/healthyweight/bmi/calculator.html). Children were classified as underweight (≤5 percentile), normal weight (6–84 percentile), overweight (85–94 percentile), and obese (≥95 percentile) as defined on the CDC website.

### 2.4. Saliva Collection

Whole saliva (5 mL) was collected from the mothers while the saliva flow was stimulated by chewing on a 1 g piece of paraffin. From the children, saliva was collected for up to 2 min using a specimen suction device (Unomedical, Malmö, Sweden), modified by Lif Holgerson from Noakes et al., 2007 [21], and attached to a slight vacuum from the dental unit. The saliva was collected into sterile, ice-chilled test tubes, which were immediately capped after collection completion and stored on ice until transferred to a −80 °C freezer within 4 h.

### 2.5. DNA Extraction

DNA was extracted using the GenElute Bacterial genomic DNA kit (Sigma-Aldrich Co., Stockholm, Sweden) from 400 μL of saliva as described earlier [22]. Briefly, after 5 min spinning at 13,000 rpm, cells were lysed in buffer with lysozyme and mutanolysin and treated with RNase and Proteinase K. Released DNA was purified, washed, and eluted, and the quality was estimated using a NanoDrop 1000 Spectrophotometer (Thermo Fisher Scientific, Uppsala, Sweden) and the quantity by the Qubit 4 Fluorometer (Invitrogen, Thermo Fisher Scientific, Waltham, MA, USA).

### 2.6. Selection of Taste- and Food Preference-Related SNPs and Sequencing

SNP variants (n = 133 in 20 genes) reported to be associated with taste, eating preferences, or reported food intakes in at least two publications were selected for the present screening. The list of selected SNPs, sequencing results, underlying references, and proposed taste associations is listed in Appendix A**.** SNP genotyping was done by MALDI-TOF (Matrix Assisted Laser Desorption-Ionisation-Time Of Flight) analysis, performed on the MassARRAY Platform from Agena (www.agenabio.com) at the Mutation Analysis Facility at Karolinska University Hospital (www.maf.ki.se). Validation, concordance, and quality control analyses were performed by the facility according to in-house standard protocols.

### 2.7. Caries Assessment

Information on the mothers’ dental status was obtained from dental electronic charts stored at Region Västerbotten, Sweden. The number of surfaces with treated caries, fillings, and missing surfaces (DMFS) was calculated, excluding 3rd molars. Teeth missing due to caries or with a crown were imputed as having 4 caries-affected surfaces for incisors and 5 surfaces for premolars and molars.

### 2.8. Data Handling and Statistical Analyses

The primary outcomes in the present study were mothers’ weekly intakes and preferences for sweet foods, weekly sweet food intake in 19 and 35 months old children, and weekly intake of sucrose (grammes/week) in 35 months old children. The exposure variables were (a) all SNP variants (for analysis of the mother’s diet) and (b) the subset of maternal SNP variants associated with the mother’s diet in the present study (for analysis of the children’s diet). One secondary objective involved testing the stability of preference and intake scores from sweet-, bitter-, sour-, and umami-tasting foods, which were assessed as the correlations of estimates from FFQ1 and FFQ2. Another secondary outcome was to compare body mass index (BMI) and caries prevalence for the SNP variants associated with sweet preference and intake in the mothers.

Univariate measures, such as mean, 95 or 99% CIs, and proportions (%), were used to describe group characteristics. Adjustments for age or anthropometric measures were performed as indicated in the text or tables using general linear modelling (glm). Associations were estimated as Pearson correlations and group differences by ANOVA in the glm modelling.

SNPs were excluded if the genotyping success rate was less than 70%, the minor allele frequency was <0.01 or Hardy–Weinberg equilibrium failed (*p* < 0.01). Associations between the primary and secondary outcomes and SNP polymorphisms were evaluated for (i) additive (AD; AA-Aa-aa), (ii) dominant major allele (D1; AA + Aa vs. aa), and (iii) dominant minor allele (D2; AA vs. Aa + aa) models using linear regression. The models included anthropometric measures to account for potential BMI-related errors in diet reporting. Age was included in models with BMI or caries as the outcome. *p*-values were adjusted for multiple SNP comparisons and outcomes using the false discovery rate (q-value < 0.05). This strategy is conservative (since all variants are anticipated a priori to be associated with dietary traits), therefore results with *p*-values < 0.05 are also presented, as are effect sizes and 95% CIs.

The software Stata 17 (StataCorp LLC, College Station, Texas) and R Studio were used for data inference (RStudio Team (2020)). RStudio: Integrated Development for R. RStudio, PBC, Boston, MA URL http://www.rstudio.com/.

## 3. Results

### 3.1. Participant Characteristics

Of the 258 eligible mothers, 251 contributed diet questionnaire data from the NorthPop database, 187 saliva for DNA extraction, and 142 information obtained by the taste preference questionnaire (Appendix A). Study participant characteristics are presented in Table 1.

### 3.2. SNP Genotyping

Eleven (8%) of the SNPs were excluded due to a call rate of less than 70% or a minor allele frequency (MAF) of zero or near zero (0.003), leaving 122 SNPs in 70 haploblocks and 20 genes for analyses. All accepted SNPs were in Hardy–Weinberg equilibrium (all *p* > 0.003). A complete list of the included SNP (rs-numbers), positions, and other descriptive information is presented in Appendix A.

### 3.3. Mothers’ Reported Intake and Preferences for Sweet and Other Taste-Representing Foods

Of the 251 mothers who had responded to an FFQ during pregnancy (FFQ1) or when the child was 24 months old (FFQ2), 241 had a recording from the first occasion and 188 from the latter occasion (Table 1). Reported intake frequencies of foods classified as sweet-, bitter-, sour-, and umami-tasting (Appendix A) were transformed into intakes per week. The sum of weekly intake frequencies in the respective taste groups was aggregated into sweet, bitter, sour, and umami intake scores. The numbers of foods used in the respective scores were eight, six, five, and four (Appendix A). Less than 2% of the FFQ questions used to aggregate these scores were left unanswered. These missing answers were considered null intakes. There was a high agreement between the number of weekly sweet intakes the mothers recorded by FFQ1 and FFQ2, i.e., nearly three years apart (r = 0.80, *p* < 0.001; Figure 1A). For the other three taste groups, the agreements between the two recordings were somewhat lower, though statistically significant, i.e., for weekly intake of bitter foods (r = 0.55, *p* < 0.001; Figure 1B), sour foods (r = 0.34, *p* < 0.001; Figure 1C), and umami-tasting foods (r = 0.46, *p* < 0.001; Figure 1D). A summary of these correlations is seen in Figure 2B.

From the taste preference questionnaire, the scores for six questions (Appendix A) were aggregated to a sweet taste score with values ranging from 18 to 34 and a median value of 27 (Figure 2A). Single missing values were inputted with the group median value for the specific food item. Corresponding scores were calculated for bitter taste (eight foods), sour taste (six foods), and umami taste (five foods). Some additional questions asked for bitter foods were excluded since the proportion of missing values was unacceptable (>5% missing responses). The distributions of the bitter, sour, and umami scores are found in Figure 2A. The summary scores for a preference for sweet-tasting foods were positively correlated with the scores for umami preference (r = 0.20, *p* = 0.020) and weekly, though not statistically significant, reported intakes of sweet foods (Figure 2B). Further, a preference for bitter-tasting foods was negatively correlated with intake of sweet foods (r = −0.26, *p* < 0.004), positively correlated with intake of bitter foods (r = −0.43/0.48 (FFQ1/FFQ2), *p* < 0.001), and with reporting a preference for sour-tasting foods (r = 0.51, *p* < 0.001) (Figure 2B). An overview of the various correlation coefficients is displayed in the heatmap in Figure 2B.

### 3.4. Associations between SNPs and Taste Preference and Food Intake in the Mothers

In total, 122 SNPs were assessed for association with sweet food preference scores and reported intake by FFQ1 or FFQ2. Prior to correction for multiple comparisons, there was nominal evidence of association (*p* < 0.05) for one or more outcomes in one or more of the tested models at 32 SNPs (Appendix A). These were in 22 haploblocks in 11 genes. Four SNPs displayed significant effects, with the sweet taste preference score as the outcome and one of the FFQ-based intakes (Appendix A). These were rs6577541 (*CA6* gene encoding the carbonic anhydrase 6 protein), rs7513755 (*TAS1R2* gene encoding the taste receptor type 1 member 2 protein, which is part of the dimeric sweet taste receptor), rs34162196 (*OR10G3* gene encoding the olfactory receptor 10G3 protein), and rs3785368 (*SCNN1B gene* encoding the β subunit of the epithelial sodium channel). Notably, both rs34162196 and rs35260863 in the same haploblock of the *OR10G3* gene were significantly associated with a higher preference for sweet foods but only rs34162196 was significantly associated with weekly intake (Appendix A). A graphic illustration of the associations (β-coefficient with 95% CI) for the 11 genes and the best-fitting SNPs versus the three outcomes, i.e., preference for sweet-tasting foods, weekly intake of sweet foods by FFQ1, and weekly intake of sweet foods by FFQ2, is found in the forest plot in Appendix A.

Following correction for multiple testing (n = 122 tests), two associations (both with sweet preference as the outcome and with BMI adjustment) remained significant (q < 0.05), i.e., the D2 model for rs7513755 in the *TAS1R2* gene (major allele T with an allele frequency of 0.72 and minor allele G of 0.28), and the D2 model for rs34162196 in the *OR10G3* gene (major allele C with an allele frequency of 0.92 and minor allele T of 0.08).

A comparison of sugary food traits in the mothers revealed higher sweet preference scores (*p* < 0.001), weekly intakes by FFQ2 (*p* = 0.044), and BMI (*p* = 0.049) for those with the minor T allele in the rs34162196 SNP in the *OR10G3* gene than those with CC (Table 2). No significant difference was seen for caries prevalence (Table 2).

Similarly, mothers with the G allele in the rs7513755 SNP in the *TAS1R2* gene reported higher intakes of sweet foods as recorded by both FFQ1 and FFQ2, but not a preference for sweet foods (Table 3). Those with the major allele G in rs3785368 in the *SCNN1B* gene had significantly higher sweet intake by FFQ2 only (Appendix A), whereas none of the sweet food measures differed between those with the C allele versus GG in rs6577541 (Appendix A). No other association was found for any of these four SNPs.

### 3.5. Child Feeding in Relation to Their Mothers’ Genetic Association with Sweet Preference

The number and anthropometric measures of children for whom sweet intakes and estimated sucrose intake were compared by their maternal rs34162196, rs7513755, rs3785368, and rs6577541 variants are presented in Table 4.

The distributions of weekly intakes of sweet products when the children were 19 and 35 months old and the estimated sucrose intake at 35 months of age are shown in Figure 3A. Correlation analysis between the mother’s and children’s various diet scores revealed notable correlations between sweet intakes in children (19 months) and mothers (FFQ1) (Figure 3B).

Children born to mothers with a T allele in the rs34162196 SNP in the *OR10G3* gene were given higher amounts (grammes) of sucrose per week compared to mothers with C alleles only (unadjusted intake of 232 compared to 158 g, (*p* = 0.021). Adjusting for mothers’ BMI and children’s BMI to account for potential diet reporting bias strengthened the difference slightly (238 versus 154 g per week, *p* = 0.01) (Table 5). In line with this, BMI-adjusted weekly intakes of sweet foods were also more frequent in children of mothers with a T allele (compared to the C allele) in the rs34162196 SNP at 35 months of age (11.0 versus 7.5 intakes, *p* = 0.022) and 19 months of age (5.4 versus 3.6 intakes, *p* = 0.046). Neither maternal rs7513755 nor maternal rs3785368 or rs6577541 variations were associated with differences in weekly intakes of sucrose or sweet foods in the children (Table 5, Appendix A).

As an attempt to distinguish associations in relation to carriage of the T allele in rs34162196, we defined groups where neither the mother nor the child carried the T allele: the mother (but not the child), the child (but not the mother), or both mother and child, and compared the children’s sucrose intake. As seen in Figure 3C,D, there was a continuous increase in the child’s sucrose intake, indicating enhancement of the association if both carry the effect allele (*p*-value among groups < 0.001 and linear trend *p* < 0.001). Including the four groups in a linear regression model yielded a significant effect of the T allele in rs34162196 even if bootstrapping for 1000 times (unstandardised β-coefficient = 51.4, *p* = 0.002).

## 4. Discussion

The present study found that the T allele in rs34162196 in the *OR10G3* gene is linked to a higher preference for sweet-tasting foods as well as aggregated intake of sweet foods and BMI. Children born to mothers with the T allele in rs34162196 were given more sucrose and more sweet treats. Further, variations in taste preference and food intake and associations with several previously reported food-associated SNPs were confirmed, such as in the *CA6, SCNN1B*, and *TAS* gene families.

The Olfactory Receptor (OR) gene family encodes a set of G-protein-coupled receptor proteins, including the *OR10G3* gene ciphering the olfactory receptor 10G3 protein. A systematic evaluation of ORs in mammals has found OR polymorphisms to link to dietary type (e.g., being herbivore, insectivore, or frugivore) [18]. In this perspective, it is interesting to note that intake of pieces of fresh fruit (rs34162196) and dried fruit (rs35260863) [11] and our present finding of a more general preference for sweet foods relate to genetic variation in *OR10G3* gene-associated SNPs. We found that both rs34162196 and rs35260863 correlated strongly with the preference for sweet-tasting products and consumption, with the parallel associations being expected since these SNPs are in the same haploblock. This is in keeping with previously reported associations between these variants and diet type [23] and specific food items, i.e., intensely sweet (around 70% glucose/fructose/sucrose) dried fruit, and fresh fruits with a mix of sweet and sour tastes [11]. The ability of humans to distinguish between edible or poisonous foods and energy-rich (e.g., sugar-containing) foods through taste and smell sensations has been crucial for survival in ancient days, but today such preference may potentially predispose to sugar-associated diseases like obesity and dental caries. The rs34162196 variant is located within a protein-coding region of OR10G3, and the T allele is annotated as a missense variant in the dbSNP database (https://www.ncbi.nlm.nih.gov/snp). The rs35260863 variant is located in a non-coding region but highly correlated with the rs34162196 variant (r2 = 0.97 in the haploreg v4.1 database https://pubs.broadinstitute.org/mammals/haploreg/haploreg.php), meaning both variants have tag variation in the OR10G3 transcript.

A key finding in the present study was that children of mothers with the T allele in rs34162196 were given more sweet treats and a total amount of sucrose per week. Since the genotype of the child correlates 0.5 with that of the mother, it may be questioned if this is driven by the mother’s preference for sweetness from her T allele or if it is driven by the child’s own preference. As an attempt to distinguish this, we created four groups for the different combinations of T allele carriage in rs34162196, i.e., neither the mother nor the child, the mother only, the child only, or both the mother and child carried the T allele. Three of the groups were small, especially the group where both the mother and child had the T allele, meaning the results of this analysis require cautious interpretation. This analysis suggested that children in the group where both mother and child had the T allele had the highest sucrose intake, while children in the group where only the mothers had the T allele had the second highest sucrose intake. This would suggest that the mother’s genetic status has a stronger effect on food selection in early childhood than the child’s status, i.e., the “mother as doorkeeper” concept, but replication in larger studies is needed to confirm this.

In the present study group, women having the T allele in rs34162196 in the *OR10G3* gene had a higher mean BMI than those with two C alleles, and, though not statistically significant, the proportion of obese mothers was nearly twice as high in the former group (20.0 vs. 11.6%). One possibility is that the T allele in rs34162196 and the C allele in rs35260863 drive patterns of food intake that increase the risk of being overweight. In support of this, these variants are reportedly associated with less intake of fruits (fresh and dried) and higher intake of bread and biscuit-type cereals (e.g., Weetabix) in the UK Biobank (https://gwas.mrcieu.ac.uk/phewas/), as well as increased intake of sugary foods (mainly treats and snacks) in the present study. Sugary foods, and especially sucrose-sweetened soft drinks in combination with high fat intake, have been linked to high BMI commonly seen in Western diet studies and hedonic eating patterns, with a possible mechanism involving imbalance in energy intake and expenditure or through gut microbiota dysbiosis in such eating patterns [24,25]. We note, however, that BMI is a complex trait, and there may be other mechanisms other than dietary preference mediated by the *OR10G3* gene that could associate with BMI.

The tested polymorphisms were not associated with dental caries in this study, which was potentially unexpected since caries is the only disease where sucrose has been proven to have a causal effect [26]. However, caries is a complex disease where concerted heritability is found to exceed 50%, but single SNPS in GWAS studies is less than 0.1% (see supplementary information in Haworth et al., (2020) [27]), which is too small to be detectable in the present study. In addition, the women in the present study group were born between the years 1980 and 2000 and grew up with regular (usually annual) dental care with a focus on preventive measures such as oral hygiene, fluoride use, and sugar reduction counselling. Thus, this generation in Sweden has among the lowest caries prevalence and variance in the world, which may reduce the ability to detect associations with caries traits.

The present study supports the concept that individual dietary habits are, at large, stable over time [28], i.e., moderate to high correlations were seen for intakes for all four taste categories reported when the women were in the third trimester and when the children were around 2 years old. Further, a higher reported intake of sweet foods was associated with less preference for bitter foods, whereas higher preference scores for sweet correlated positively with the scores for umami-tasting foods. The taste receptors for sweet and umami share a common unit in their respective dimeric receptor proteins, which hypothetically may be an explanation of a relationship between the perception of umami and sweet taste at the molecular level. Such an association has been suggested in animal studies and in human umami hypotasters who have significantly lower umami and lower sweet taste perception [29,30], and combining sweet and umami in foods and meals is appreciated, but as the associations here are weak, further studies need to confirm or reject the finding.

The strengths of the present study relate to the comparably wide panel of SNPs evaluated in mother-child pairs and the fact that SNP polymorphisms could be compared for diet exposures collected at two different time points for both the mothers and the children. Nevertheless, some weaknesses should be considered when the results are interpreted. First, the number of included participants restricts the power to detect associations with SNPs with comparably high allele frequencies. This is reflected in the borderline significance seen for most findings and the fact that only the two most influential variants remained significant after FDR correction. However, since all variants were anticipated a priori to be associated with dietary traits and, in fact, were in large part confirmatory of previous findings, we argue that *p*-values < 0.05 are indicative of group differences in effect sizes. Further, the lack of DNA and dietary information excluding the fathers from the analyses and the risk of a systematic error in dietary recordings should be recognised. Though a comprehensive validity study is yet to be published, a pilot study in a small sub-group of mothers in NorthPop has indicated a relative validity similar to other FFQs [31,32]. Accordingly, information on estimated energy intakes was not available to allow for energy standardisation of reported intakes through, e.g., the residual method. However, body mass index, which is one well-known risk factor for systematic error in diet recordings, was included as a covariate in all models with diet as the outcome. Possibly, the limitation to ≤12 food items in any intake score also limits the extent of misreporting. Further, like all genetic association studies, the study is potentially susceptible to the effects of population stratification, but we do not think it explains the associations here because 94.1% of the women were from countries of predominantly European ancestry (Sweden n = 174, UK n = 2).

## 5. Conclusions

In conclusion, minor allele variation in mothers’ rs34162196 was strongly associated with their sweet preference and sugar intake in their children. The rs34162196 might be a candidate for a genetic score for sweet intake to complement self-reported intakes, and knowledge on innately driven cravings for sweet may be helpful in individualised diet counselling for both the mother’s and child’s health.

## Figures and Tables

**Figure 1 nutrients-15-02565-f001:**
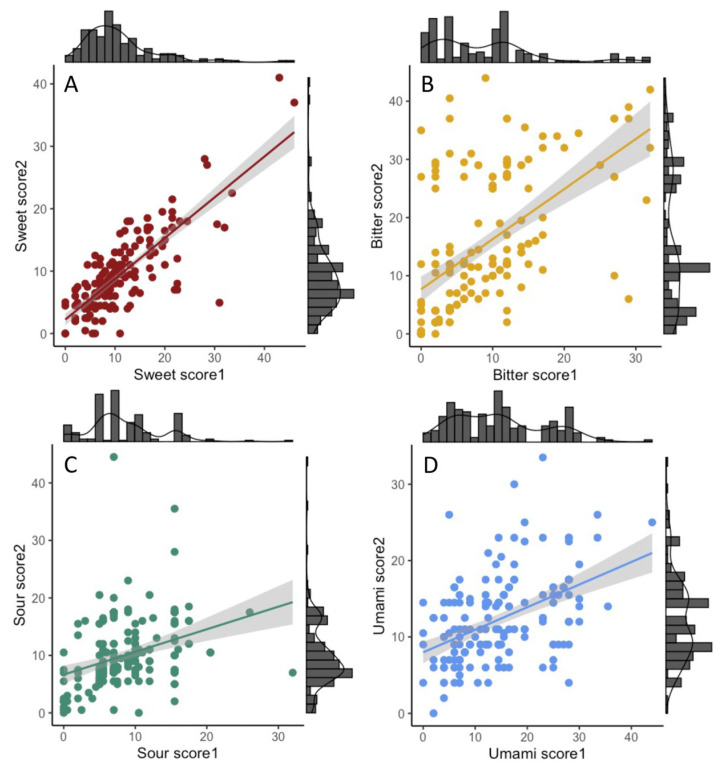
Scatterplots with regression lines with 95% confidence intervals for (**A**) summary scores for sweet-tasting foods, (**B**) bitter-tasting foods, (**C**) sour-tasting foods, and (**D**) umami-tasting foods. Scores numbered 1 are from the FFQ answered when the mothers were in the 3rd trimester (FFQ1), and those numbered 2 are from the FFQ when the children were 24 months old (FFQ2). Adjacent to the scatter plots are the respective histograms with density lines.

**Figure 2 nutrients-15-02565-f002:**
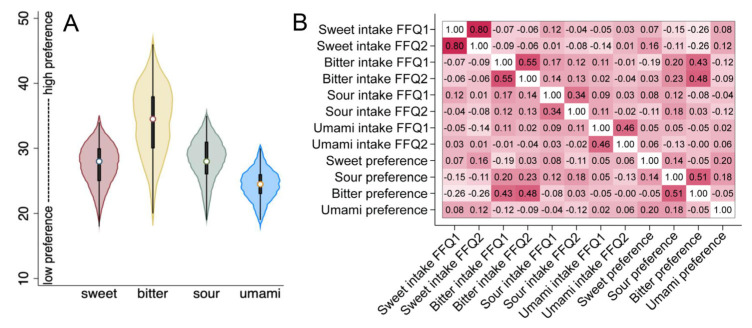
(**A**) Violin plots showing the summary scores for the reported liking of marker foods for sweet, bitter, sour, and umami tastes. The scores were approximately normally distributed, with greater variance in the score for bitter taste preference than the other scores. (**B**) A heatmap illustrating correlations between the diet scores estimated by participating mothers. The numbers in the cells are the correlation coefficients. Variables labelled FFQ1 refer to recordings in pregnancy and those labelled FFQ2 when the children were 24 months old.

**Figure 3 nutrients-15-02565-f003:**
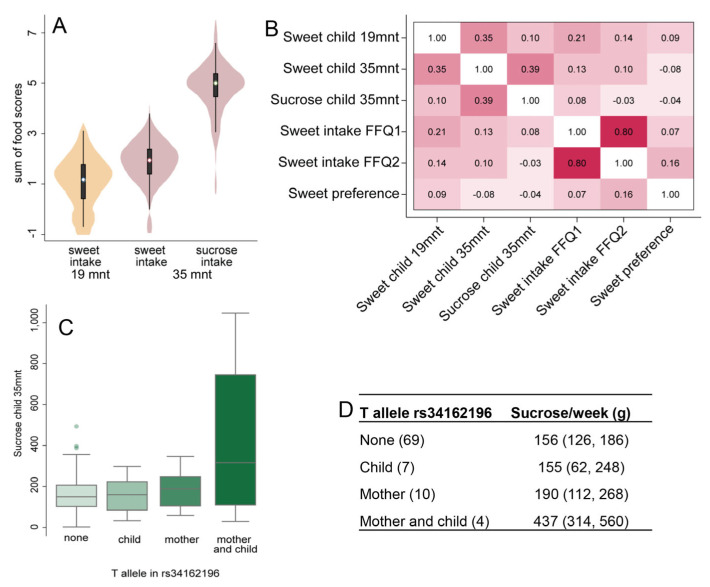
(**A**) Violin plots illustrating the distribution of summary scores from children’s intakes per week of sweet treats at ages 19 and 35 months, respectively, and weekly total sucrose intake at the latter age. The data are ln transformed for a better fit in the figure. (**B**) A heatmap illustrating associations between the children’s sweet/sucrose intake and the mothers’ sweet intake and preference. (**C**) Box plots showing weekly sucrose intake in 35 months old children by presence of the T allele in rs34162196 for pairs of mother and child. The numbers in parenthesis refer to the number in each group, with (**D**) giving the mean (95% CI) values for sucrose intake as grammes per week for the same groups (*p* < 0.001).

**Table 1 nutrients-15-02565-t001:** Characteristics of the 251 mothers who contributed information from at least one diet screening (FFQ1 and/or FFQ2).

Variable	Measurement
Age (years mean, 95% CI; min-max) ^1^	31.1 (30.5, 31.6; 19.45)
BMI (kg/m^2^, mean, 95% CI)	24.3 (23.8, 24.8)
Underweight (BMI < 20 kg/m^2^), %	8.1
Normal weight (BMI 20 ≤ 25 kg/m^2^), %	54.3
Overweight (BMI 25 ≤ 29 kg/m^2^), %	24.8
Obese (BMI ≥ 30 kg/m^2^), %	10.1
University or college education, %	68.2
Maternal FFQ1 in the 3rd trimester, n (%)	241 (96.0)
Maternal FFQ2 when the child is 24 months old, n (%)	188 (74.9)
Taste preference questionnaire, n (%)	142 (56.7)
Saliva sample for DNA extraction, n (%)	187 (74.5)
Oral examination, n	208
Number of teeth (mean, 95% CI; min–max)	27.3 (27.1, 27.5; 15–28)
Caries prevalence (mean, 95% CI; min–max) ^2^	11.2 (9.8, 12.6; 0–59)

^1^ Age at childbirth. ^2^ Sum of Decayed, Missing, and Filled tooth surfaces in permanent teeth (DMFS). 3rd molars were excluded, and means were adjusted for age at the oral examination.

**Table 2 nutrients-15-02565-t002:** Phenotype comparison in mothers by rs34162196 (in the *OR10G3* gene) variation. Sequencing had failed in three samples, leaving 184 mothers in the model.

	Major Allele CC (n = 148)	Minor Allele CT/TC/TT (n = 36)	*p*- Value
Diet, mean (95% CI) ^1^			
Sweet preference score	26.9 (26.3, 27.5)	29.5 (28.3, 30.7)	<0.001
Bitter preference score	34.4 (33.3, 35.4)	32.4 (30.2, 34.7)	0.128
Sour preference score	29.1 (27.5, 28.7)	28.7 (27.4, 30.0)	0.395
Umami preference score	24.2 (23.8, 24.6)	24.9 (24.0, 25.9)	0.156
Sweet food intake/week (FFQ1)	10.4 (9.2, 11.7)	11.5 (8.9, 14.2)	0.489
Sweet food intake/week (FFQ2)	9.1 (8.1, 10.2)	11.8 (9.4, 14.1)	0.044
Bitter food intake/week (FFQ1)	8.8 (7.6, 9.9)	7.7 (5.2, 10.0)	0.413
Bitter food intake/week (FFQ2)	15.2 (13.0, 17.3)	14.7 (9.8, 19.5)	0.854
Sour food intake/week (FFQ1)	8.4 (7.6, 9.2)	7.6 (5.9, 9.3)	0.380
Sour food intake/week (FFQ2)	10.3 (9.3, 11.4)	8.3 (5.9, 10.7)	0.123
Umami food intake/week (FFQ1)	13.5 (12.1, 15.0)	17.0 (13.9, 20.0)	0.045
Umami food intake/week (FFQ2)	12.1 (11.1, 13.2)	12.0 (9.7, 14.3)	0.918
**Secondary outcomes, mean (95% CI)**			
BMI, kg/m^2^	24.1 (23.4, 24.8)	25.6 (24.2, 27.0)	0.049
Proportion overweight, %	21.9	25.7	0.314 ^2^
Proportion obese, %	11.6	20.0	
DMFS ^3^	10.7 (8.9, 12.6)	12.6 (9.0, 16.1)	0.364

^1^ Means and 95% CI adjusted for mothers’ BMI. ^2^ Pearson chi-square from the cross-tabulation with the three body-weight levels. ^3^ Sum of Decayed, Missing, and Filled permanent tooth surfaces. Means and 95% CI adjusted for age at the oral examination.

**Table 3 nutrients-15-02565-t003:** Phenotype comparison in mothers by rs7513755 (TAS1R2 gene) variation. Sequencing was successful in all samples.

	Major Allele TT (n = 156)	Minor Allele TG/GT/GG (n = 31)	*p*- Value
**Diet, mean (95% CI) ^1^**			
Sweet preference score	27.5 (26.9, 28.1)	26.8 (25.5, 28.1)	0.343
Bitter preference score	34.3 (33.2, 35.3)	32.8 (30.5, 35.1)	0.252
Sour preference score	28.3 (27.7, 28.9)	27.9 (26.5, 29.2)	0.536
Umami preference score	24.4 (24.0, 24.8)	24.0 (23.1, 24.9)	0.451
Sweet food intake/week (FFQ1)	10.1 (8.9, 11.3)	13.2 (10.4, 16.0)	0.046
Sweet food intake/week (FFQ2)	9.0 (7.9, 10.0)	11.8 (9.5, 14.1)	0.031
Bitter food intake/week (FFQ1)	8.7 (7.6, 9.8)	7.5 (5.0, 10.1)	0.414
Bitter food intake/week (FFQ2)	15.8 (13.7, 17.9)	11.5 (6.7, 16.2)	0.102
Sour food intake/week (FFQ1)	8.4 (7.7, 9.1)	7.9 (6.1, 9.7)	0.642
Sour food intake/week (FFQ2)	10.2 (9.1, 11.3)	9.2 (6.7, 11.6)	0.448
Umami food intake/week (FFQ1)	13.9 (12.5, 15.3)	15.8 (12.6, 19.0)	0.277
Umami food intake/week (FFQ2)	12.0 (11.0, 13.0)	13.3 (11.0, 15.6)	0.302
**Secondary outcomes, mean (95% CI)**			
BMI kg/m^2^	24.6 (23.9, 25.2)	23.3 (21.8, 24.7)	0.112
Proportion overweight, %	24.0	16.7	0.516 ^2^
Proportion obese, %	13.6	10.0	
DMFS ^3^	11.3 (9.5, 13.1)	10.8 (6.8, 14.7)	0.808

^1^ Means and 95% CI adjusted for mothers BMI. ^2^ Pearson chi-square from the cross-tabulation with the three body-weight levels. ^3^ Sum of Decayed, Missing, and Filled permanent tooth surfaces. Means and 95% CI adjusted for age at the oral examination.

**Table 4 nutrients-15-02565-t004:** Characteristics of the children for whom sweet intakes and estimated sucrose intake were compared by maternal genetic variation in rs34162196, rs7513755, rs3785368, and rs6577541.

Variable	Measurement
Child FFQ at 19 months, n (%)	203 (80.9)
Child FFQ at 35 months, n (%)	161 (64.1)
Child BMI percentile at 35 months (mean, 95% CI)	60.3 (56.2, 64.3)
Underweight at 35 months (BMI < 5% percentile), %	3.1
Normal weight at 35 months (BMI 6 ≤ 85 percentile), %	71.5
Overweight at 35 months (BMI 85 ≤ 95 percentile), %	11.1
Obese at 35 months (BMI ≥ 95% percentile), %	14.2

**Table 5 nutrients-15-02565-t005:** Weekly sucrose and sweet food intakes in children of mothers carrying the CC allele versus not in rs34162196 and in children of mothers carrying the TT allele versus not in rs7513755.

Diet in Children (Mean (95% CI) ^1^) by Maternal rs34162196 SNP Polymorphism	Major Allele CC (n = 148)	Minor Allele CT/TC/TT (n = 36)	*p*-Value
Sucrose intake/week, 35 months old	154 (127, 181)	238 (181, 295)	0.010
Sweet food intake/week, 19 months old	3.6 (2.8, 4.3)	5.4 (3.8, 7.1)	0.046
Sweet food intake/week, 35 months old	7.5 (6.3, 8.7)	11.0 (8.3, 13.6)	0.022
**Diet in Children (Mean (95% CI) ** ^ **1** ^ **) by Maternal rs7513755 SNP Polymorphism**	**Major allele** **TT** **(n = 156)**	**Minor allele** **TG/GT/GG** **(n = 31)**	* **p** * **-** **Value**
Sucrose intake/week, 35 months old	175 (148, 201)	132 (74, 191)	0.196
Sweet food intake/week, 19 months old	3.6 (2.8, 4.4)	5.1 (3.3, 6.8)	0.135
Sweet food intake/week, 35 months old	8.2 (6.9, 9.4)	7.3 (4.6, 10.0)	0.568

^1^ Means and 95% CI adjusted for the mother’s BMI and the child’s BMI percentile.

## Data Availability

Data described in the manuscript, including the questionnaires used, will be made available upon request pending valid ethical approvals and approval by the NorthPop steering committee.

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
