# Peer review of "Genetic Preference for Sweet Taste in Mothers Associates with Mother-Child Preference and Intake"

_nutrients, 2023, doi:10.3390/nu15112565_

Round 1

Reviewer 1 Report

This is an interesting study; however, some aspects of the manuscript need to be improved.

I have some concerns, particularly as it related results presentation (see below).

Title

I suggest to revise “child’s diet profile” in the title, with a more appropriate description reflecting the results of the manuscript

Abstract,

Only result of OR10G3 SNP has been discussed. Authors should also explain results on TAS1R2 SNP

Introduction

-       Authors should introduce ionic channels involved in sour and salty taste perception

-       Line 49: correct “a bitter taste”. TAS2R genes confer the perception of a variety of bitter compound

-       Line 70-71: clarify the primary objective of the study

Materials and Methods

-       Line 149: authors state that select SNPs “associated with taste or eating preferences”. However, several SNPs associated with intake measures were included in table S3. For example, OR10G3 SNP was found to be associated with fruit intake, as authors report in the introduction.

-       Line 169-171: better specify “measures of dietary recording stability for the mothers” 

Results

I found results difficult to follow. I don’t understand because authors state that two association are significant after correction for multiple testing, but report results on 4 SNPs. I suggest to focalize results description only on the two significant SNPs and to do the following changes:

-       3.2 section could be moved in materials and method section.

-       Also taste score calculation and description reported in 3.3 section could be moved in materials and method section. 

-       Figure 2A and 4A could be removed or moved in supplementary materials

-       Figure 3 could be moved in supplementary materials. While results on rs7513755 presented in Table S5 should be showed in the main text, as for table 2.

-       Overall, results on children could be removed from table 2 and S5 and report in a separated table.

-       Figure 4D is redundant, could be removed. Number in each groups could be added in figure 4C

Moreover, I don’t see Table S1 in supplementary materials

Discussion

As for the results, a better organization of this section is needed.

Authors should discuss mainly statistically significant results. Therefore, authors should consider to modify lines 325-330.

Moreover, I don’t understand because authors discuss rs35260863 but findings on this SNP are not mentioned in results section.

Overall, the manuscript needs also careful writing editing. Below some examples:

Line 51: “in the family” should be “the family”?

Line 75: clarify “variations in a genetic preference for sweet”

Line 263: correct “sore”

Line 430: “and intake”, repeated

Reviewer 2 Report

The manuscript entitled: “Genetic preference for sweet taste in mothers associates with 2 her and her child´s diet profile” is of interest in detecting research gaps of sweet food intake and taste preferences in relation to genetic between mothers and children.

Although, the quality of the manuscript is quite high, some issues and comments are relevant to be considered and adopted in the current manuscript.

L22 change to: mothers and children or mother and child pairs.

L33 Introduction: fat perception should be included and more information on sweet taste as this was the main research focus referring to the hypothesis, especially the development of innate and acquired sweet taste (preferences).

L68-69 several studies are stated but only one references are given, present primary literature

L69 and on the other hand side they will bequest preferences too (so genes and behavior), this should be added to be more precise

L98 The FFQ1 was recorded for 12 month – so there were 2 different periods recorded – before and during pregnancy – as it is known that especially preferences change here a lot, why not only one of these periods was recalled (as it is also known that it is difficult to remember such a long period)? Please, explain and add a statement to the method section.

L100 Please explain why the scales across food/beverage from 5 to 7 level scales were mixed and not the same across all categories as it is widely done in literature? How was the amount for an item calculated?

L101-105 Was the questionnaire for the acceptance of tastes used here for the first time? Was it developed for this study – was it validated? If not, please add the reference. Why did you use a six point scale and how were the values exactly labelled? 

L110-111 The FFQ for the child seems quite different from that for the mother, why did you decide like this and what is the basis for the questionnaire? Add the appropriate literature, was it already validated?

L125 include unit with the numbers

L150 dot instead of comma?

L148 in S3 SNPs are presented that are reported to be associated with taste or eating preferences. In the hypothesis it was mentioned that only the sweet taste is investigated. Please explain, why the SNPs search to further tastes was extended? 

L190 one time ) is missing

Table 1 child relative body weight – which unit was used here? 60.3 kg maybe not for 35 month or is it the percentile for weight? Or for BMI? , also the height of children should be added into the table

L 212 it is not clear what has been aggregated. What is the outcome?  A total intake of “tasting” foods per week?

L 214 Table S1 it is not clear why the table is listed here as it is the description of cross-sectional studies? What is the relevance?

Table S2 Sweet intake/overall food intake, it is without unit (e.g. g), but what is it? Portions?

Add sample questions including the scales to the respective food items or questionnaires, it should be made somehow clearer what the questionnaires are about then all the results make more sense.

Figure 2 A -  include a labelling for the liking. E.g. at 10 low liking, at 40 high liking

Figure 2 B – it is still not clear what is meant with sweet taste- is it preference or perception, the wording should be changed here.

L263 sore should be score?

Figure 3A Preference = sweet preference? (add the word, otherwise it can be referred to overall or every other taste/food preference) and/or sweet intake for FFQ1/FFQ2 to make it already clear for the readers.

Overall the termini should be used consistent throughout the manuscript figures and tables  e.g. sweet preference or sweet preference score (then it is clear that it is without unit) and sweet food intake/week or sweet food intake or sweet intake per week, or sweet intake/week, weekly intake of sweet foods

Now there are too many different versions that makes it very complicated in terms of sensory language to read and understand the paper.

L301 P=0.087 vs. 0.0.087

Figure 4A 35 mmt vs.mnt

Figure 4B  is it the sweet taste or preference – this is unclear - same as I mentioned above

Figure 4 D is the result significant (e.g. using ANOVA)

L334 a ) is too much

L341 (18) vs. [18]

Overall, was it considered to include the fathers’ intake/preferences and alleles somehow? This might be important in addition to the mother’s ability to bequeath. A statement should be included to the discussion and considered as a weakness.

Round 2

Reviewer 1 Report

The authors have been generally responsive to my comments. 

I still have additional few concerns that should be addressed.

Abstract

Word limit should be respected.

Moreover, abstract conclusion should be also modified, considering described results (not only rs34162196)

Introduction

Lines 80-88

Primary and secondary aims remain still unclear.  

For example, authors should specify what means for maternal innate driving forces and use the correct terminology (preferences and intakes are not synonymous), 

Also the sentence “Secondary objectives included assessment of associations with other taste variants, BMI, and dental caries. The exposure variables were a panel of 133 single nucleotide polymorphisms (SNPs) in genes reported to be associated with eating preferences and intake, which were sequenced in 187 mothers” is not clear and explanatory. 

Moreover, primary and secondary aims did not match with primary and secondary outcomes reported in 2.8 section. Please clarify

Materials and Methods

Line 182: What authors mean for “the stability of preference and intake”?

Results

Section 3.5, Lines 311-316: the sentence and table legend are unclear.  “could be compared by their maternal rs34162196, rs7513755, 312 rs3785368, and rs6577541 variants” should be rewrite

In table 4, specify if child BMI percentile refers at 18 or 36 months.
